# Longitudinal ultrasound assessment of jugular venous pressure reliably detects hypervolemia: An observational study in healthy volunteers

Karin Vogt[1], David Widmer[2], Mark Kirsch[3], Laura Potasso [1,4]*

1 Department of Endocrinology, Diabetology and Metabolism, University Hospital Basel, Basel, Switzerland, 2 Intensive Care Unit, University Hospital Basel, Basel, Switzerland, 3 Division of Internal Medicine, University Hospital Basel, Basel, Switzerland, 4 University of Basel, Basel, Switzerland

* laura.potasso@usb.ch

## Abstract

### Background

Ultrasound (US) assessment of the internal jugular vein (IJV) to measure ultrasound jugular venous pressure (uJVP) has been proposed as a promising non-invasive tool for evaluating hydration status. However, a standardized method for uJVP measurement is currently lacking.

### Methods

This cross-sectional study compared 4 previously described IJV-US methods in healthy, euvolemic volunteers. Methods 1 and 2 use transverse view, measuring uJVP where IJV is (1)smaller than common carotid artery or (2)collapsed throughout the respiratory cycle. Methods 3a/b use longitudinal view, measuring at the tip(3a) or base(3b) of the IJV tapering portion. US assessment was conducted by 2 independent, blinded investigators at individualized head-of-bed elevation angle according to standardized procedures. Endpoints were: a)percentage of participants with measurements differing >1 standard deviation(SD) from the previously described mean $(6.5 \pm 1.5 \, cmH_2O)$ for euvolemic patients, b)reproducibility and c)feasibility.

### Results

30 participants(50% females) were included. Median uJVP(IQR) was 3.8 $cmH_2O$(3.0–4.0), 4.5 $cmH_2O$(3.8–5.3), 3.9 $cmH_2O$(3.3–4.5), and 3.3 $cmH_2O$(2.8–3.5) for methods 1-3b. uJVP differed >1SD in 17/30(56.7%), 3/14(21.4%), 14/29(51.7%) and 25/30available assessment pairs(83.3%), respectively. Reproducibility for methods 1-3b was 18/30(60.0%), 7/30(23.3%), 19/30(63.3%) and 25/30 participants(83.3%). Assessment was feasible in 30/30 participants(100%) for methods 1 and 3b, 29/30(97%) for method 3a and 14/30(47%) for method 2.

**Data availability statement:** All relevant data are within the manuscript. In addition, we added data about the measurements as supplementary information, completely anonymised. Consent for publication of raw data not obtained but dataset is fully anonymous.

**Funding:** This study was financially supported by the the Margot and Erich Goldschmidt & Peter René Jacobson-Foundation in the form of a research grant (122023) received by LP. No additional external funding was received for this study. The funder had no role in study design, data collection and analysis, decision to publish, or preparation of the manuscript.

**Competing interests:** The authors have declared that no competing interests exist.

**Abbreviations:** CCA, common carotid artery; CVP, central venous pressure; FAS, full analysis set; IJV, internal jugular vein; IQR, interquartile range; IVC, inferior vena cava; POCUS, point-of-care ultrasound; RAP, right atrial pressure; SD, standard deviation; uJVP, ultrasound jugular venous pressure; US, ultrasound; VAS, visual analogue scale; VExUS, venous excess ultrasound.

## Conclusion

The longitudinal method measuring uJVP at the base of tapering portion showed the highest reproducibility and feasibility. Overall, measured uJVP values were consistently within the upper threshold for normal central venous pressure, but lower than the previously described values, supporting the use of uJVP for detecting hypervolemia, but not hypovolemia.

## 1. Introduction

Hydration status assessment is essential in many clinical conditions, such as dyspnea, electrolyte imbalances, hypotension, sepsis, heart, liver and kidney failure as well as in the perioperative setting. However, it represents a challenge for the physician, as to date no validated, effective, non-invasive global hydration status assessment protocol exist [1–3]. Ultrasound (US) is often employed for non-invasive hydration status evaluation [1,2,4–11]. Inferior vena cava (IVC) US [12,13] is the most widely used method, supplemented by lung US [14,15], portal/hepatic/renal vein flow assessment (venous excess US, VExUS) [16,17], and cardiac US, depending on clinical suspicion and technical applicability [1,10,11]. Yet, abdominal US assessments can be challenging in the presence of different factors such as obesity, wound dressings or bowel gas [18]. Furthermore, VExUS and cardiac US are time-consuming and complex, limiting their feasibility in ward or emergency settings because they require specially trained physicians [19–21].

Clinical examination of the neck veins has long been a fundamental component of bedside hydration status assessment [22], though the external jugular veins are not always reliably visualized [23,24]. Internal jugular vein (IJV) US and the estimation of US jugular venous pressure (uJVP) as a surrogate for the invasively measured central venous pressure (CVP) and right atrial pressure (RAP) in use for hydration status assessment, have emerged as promising non-invasive alternatives, given their feasibility and ease of learning [25–27]. However, despite a growing body of literature on IJV-US, there is no consensus on the methodology or standardized definition of uJVP.

To date, there are 4 commonly used approaches to visualize and measure uJVP, of which 2 use transverse and 2 longitudinal view [28–36]. The primary objective of this methodological, praxis-oriented study was therefore to evaluate the accuracy of the four previously described uJVP measurement methods. Additionally, we analyzed the reproducibility and feasibility of these methods.

## 2. Methods

### 2.1. Trial design and oversight

This investigator-initiated, single-center, observational cross-sectional study with blinded assessment was conducted in healthy adult volunteers. The study took place from 22nd October 2024–3rd December 2024 at University Hospital of Basel, Switzerland, and was funded by a research grant from the Margot and Erich

Goldschmidt & Peter René Jacobson-Foundation. Ethical approval was obtained from the Ethics Committee Northwest Switzerland (EKNZ 2024−01640), and the study was registered on ClinicalTrials.gov (Identifier: NCT06706960). The trial steering committee was composed of the authors who designed and supervised the trial. Data quality for each visit was reviewed by an investigator who had not participated in that particular visit. All participants provided written informed consent before enrollment. Expenses related to the study visit were reimbursed, no financial compensation was offered.

## 2.2. Inclusion and exclusion criteria

Thirty healthy adults aged 18 years or older, 15 males and 15 females, were eligible. Exclusion criteria were selected to ensure an euvolemic study population. Individuals with conditions affecting body water homeostasis, those presenting with abnormal vital signs, or with possible confounding factors were excluded. For detailed information see Table 1.

## 2.3. Trial procedures

At the start of the visit, vital signs and fluid intake data were recorded. All participants were independently examined on the same day by two blinded examiners trained in Point-of-Care Ultrasound (POCUS). The examiners consisted of 2 internal medicine residents, 1 senior internal medicine physician, and 1 endocrinologist, all following a standardized protocol. To ensure consistency, the examiners completed two 2-hour training sessions on the protocol before the study. US assessments were conducted using the linear transducer of a handheld GE° VScan Air CL probe, connected to an Apple° iPad. The right IJV was scanned with participants positioned on a bed in a semi-recumbent posture (Fig 1), head held in a neutral or slightly leftward-turned position (no more than 30°).

The head-of-bed elevation angle was individually adjusted using a standardized procedure under US guidance before initiating the examination and was kept consistent for both investigators. The goal was to visualize uJVP in the mid-third of the neck to minimize confounding factors such as proximity to the confluence with the subclavian vein or the presence of venous valves, as previously described in the literature [28,37,38].

**Table 1. Inclusion and exclusion criteria.**

| |
|---|
| <u>Inclusion criteria:</u><br>**Healthy adults aged 18 years or older** |
| <u>Exclusion criteria:</u><br>1. Major cardiovascular event in the last 3 months<br>2. Pregnant or lactating women<br>3. Heart failure of any grade<br>4. Kidney failure<br>5. Thrombosis of Internal Jugular Vein<br>6. Atrial Fibrillation<br>7. Valves impairment<br>8. Uncontrolled Diabetes mellitus<br>9. Uncontrolled Diabetes insipidus/arginine vasopressin resistance or deficiency<br>10. Respiratory Distress of any grade<br>11. Signs/Symptoms of volume loss (diarrhea, vomiting, bleeding) in the past 3 days<br>12. Medication: Angiotensin-converting enzyme inhibitors, Angiotensin receptor blockers, any diuretic therapy<br>13. Inability to follow procedures or insufficient knowledge of project language<br>14. Inability to give consent<br>15. Abnormal vital signs: tachycardia > 90/min, systolic blood pressure < 90 mmHg |
| Exclusion criteria 15 checked by the staff member.<br>Exclusion criteria 13–14 at discretion of staff member.<br>Exclusion criteria 1–12 evaluated by asking participants about their medical history. |

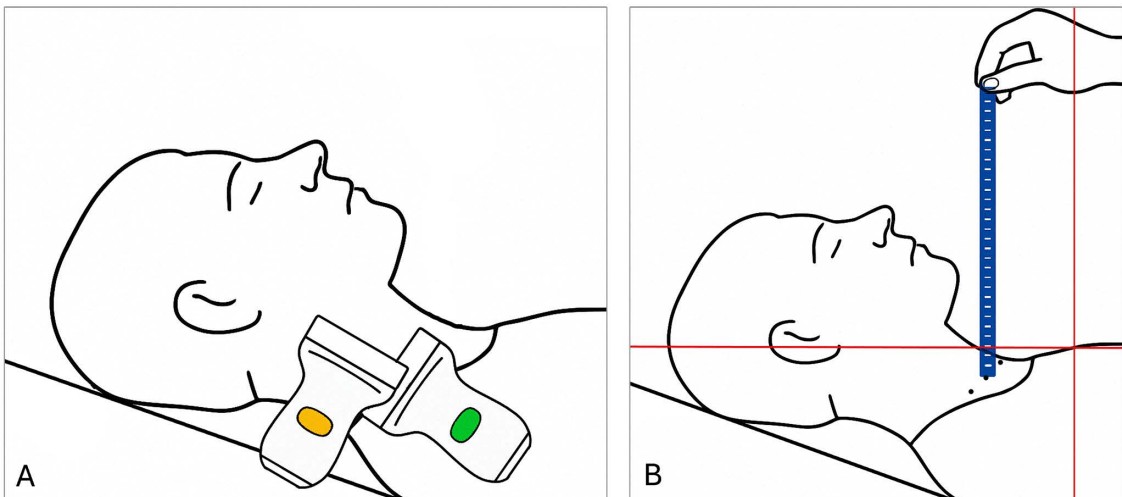

**Fig 1. Internal jugular vein assessment steps.** A: graphic showing participant and probe positioning (transverse assessments in green, longitudinal assessments in yellow); B: graphic showing the technique applied to measure internal jugular vein (IJV) height in cmH$_2$O.

uJVP was measured in centimeters of water (cmH$_2$O). For each method, IJV height was marked on the skin using a skin marker. Sternal angle served as the reference point for measurements (above= positive values, below= negative values), a horizontal reference line was projected onto the body surface. Measurements were taken vertically and rounded to the nearest 0.5 cm. In accordance with standard practice, 5 cmH$_2$O (the assumed constant distance from right atrium to sternal angle) were added to the measured IJV height to calculate uJVP [23]. The time from the start of the examination to IJV height skin mark was measured and recorded for each method. Acceptance of each US protocol by the participant was assessed using a visual analogue scale (VAS, 10 = best). Predefined US images were stored and used by the data safety monitoring board members to verify the results. To minimize potential bias, the order of measurement methods was alternated between participants and kept in the same sequence for both investigators. All methods began with the ultrasound probe positioned at the clavicle, scanning upwards.

The ultrasound methods for measuring IJV height were as follows (Fig 2):

1) Transverse view, measurement at the point where IJV is smaller than CCA throughout the entire respiratory cycle (called method 1)

2) Transverse view, measurement at the point where IJV is completely collapsed throughout the entire respiratory cycle (method 2)

3) Longitudinal view, measurement at either the tip (method 3a) or the base (method 3b) of the IJV tapering portion

   Because identical probe positioning was used for the last 2 methods, they were summarized as "method 3, a + b".

## 2.4. Endpoints

The primary aim of the study was to assess the accuracy and reliability of the current used uJVP assessment methods by comparing the assessment results with the existing normal values reported in the literature. Therefore, the primary endpoint was set at the proportion of participants whose measurements differed >1 standard deviation (SD) from the expected uJVP value, defined as 6.5 +/- 1.5 cmH$_2$O independent from sex for each of the 4 applied methods. This value based on common clinical practice [39] as well as previous studies. Those studies included a) a study by Socransky et

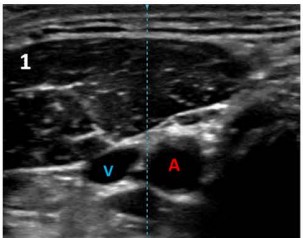 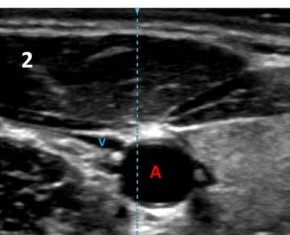 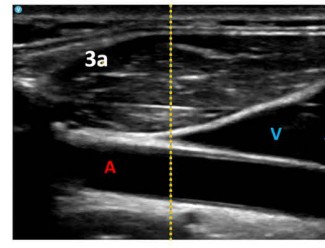 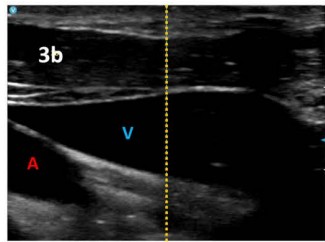

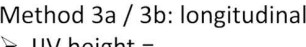

Method 1: transverse
➢ IJV height = as soon as IJV < CCA throughout entire respiratory cycle

Method 2: transverse
➢ IJV height = as soon as IJV completely collapsed throughout entire respiratory cycle

Method 3a / 3b: longitudinal
➢ IJV height =
   a) at the tip of the tapering portion (collapsing point / top of pulsation)
   b) at the base of the tapering portion (taper point)

**Fig 2. Ultrasound jugular venous pressure assessment methods.** Ultrasound images for methods 1 to 3b. IJV = internal jugular vein, A = artery, V = vein, CCA = common carotid artery.

al in supposedly euvolemic emergency department patients, defining normal uJVP using method 3b [35], b) a systematic review by Wang et al comparing different uJVP and other IJV-US cut-off values for hypo- and hypervolemia [25], and c) a study by Wang et al applying method 1 for uJVP assessment in patients scheduled for right heart catheterization [36]. The null hypothesis stated that there would be no difference compared to the expected value, i.e., that all methods will be equally accurate in measuring uJVP. The percentage was calculated by dividing the number of participants whose measurements differed by > 1 SD from the expected uJVP value by the number of participants for whom both investigators were able to measure uJVP (= number of available assessment pairs) and then multiplying the result by 100.

We performed a sensitivity analysis by investigating possible sex-related differences in the uJVP measurements.

Secondary endpoints included:

a) Reproducibility: inter-observer agreement for measurements of each method

b) Analysis of feasibility across methods

## 2.5. Statistical analysis

The sample size was determined assuming an expected uJVP mean ± SD of 6.5 ± 1.5 cmH$_2$O independent from sex [25,35,36]. To avoid bias due to measurement deviations, the study population was calculated assuming a mean in the study group of 5.5 or below or 7.5 cmH$_2$O or above. Using the equation

$$N = (SD\beta\ error +\ SD\alpha\ error\ )2/\ (population\ mean -\ mean\ of\ study\ population\ )2$$

with alpha error of 0.05 and a power of 95%, 29 participants were needed to verify the hypothesis. We rounded it at 30 to preserve the 50% sex distribution.

Data was collected in the online database Redcap®. Analysis was based on full analysis set (FAS).

We conducted a descriptive analysis. Mean and standard deviation (SD) were calculated for age, while median and interquartile range (IQR) were used for all other continuous variables. Percentages were used for binary variables. We compared uJVP values to the predefined threshold using a one-sample Wilcoxon signed-rank test. Differences between methods within the same participants were assessed using a paired Wilcoxon signed-rank test. Sex differences in uJVP values were evaluated using the Wilcoxon rank-sum test (Mann–Whitney U test). Results were visualized using box plots.

The percentage of participants differing >1 SD from the expected value was calculated. Reproducibility was calculated in percentage of participants having the same evaluation from the two independent investigators with a tolerance of maximum 1 cm, divided by the total number of participants. Additionally, Bland-Altman analysis was performed to compare the measurements obtained by the two investigators, allowing assessment of bias and limits of agreement. Two-way Intra-class Correlation Coefficient (ICC) with absolute agreement, 95% confidence intervals (CI) and p-value were reported for all methods. Feasibility was assessed on the base of a) percentage of participants in which the IJV-height visualization was possible as defined by the method, divided by the total number of participants; b) duration of each available assessment in seconds and c) acceptance of the different US-assessment protocols by all the participants based on VAS. All analyses were performed using R statistical software [40].

### 2.6. Writing process/ use of artificial intelligence

ChatGPT (GPT-4o) was used to verify grammatical accuracy. Fig 1 was generated using artificial intelligence (DALL·E 3 via ChatGPT (GPT-4o)), based on photos of the study setting.

## 3. Results

### 3.1. Participant enrollment and baseline characteristics

Thirty volunteers (15 men, 15 women) were recruited. The mean age was 33.0 years (SD±10.2), and the median BMI was 22.9 kg/m$^2$ (IQR 21.6–25.7). In total, 11 participants (36.7%) were taking medications. Most were females (n=9), with eight using oral contraceptives and one taking levothyroxine. Of the two males, one was taking ibuprofen for a minor injury, and the other a low-dose selective serotonin reuptake inhibitor (SSRI). Detailed baseline characteristics are depicted in Table 2. The head-of-bed elevation angle during assessment was 20° in 33%, 15° in 30%, 10° in 20% and 25–45° in 17% of participants.

**Table 2. Baseline characteristics of study participants.**

|  | Female | Male | Overall |
|---|---|---|---|
| n (%) | 15 (50) | 15 (50) | 30 (100) |
| Age<br>years (mean, SD) | 33.0 (9.9) | 33.0 (10.8) | 33.0 (10.2) |
| Body mass index BMI<br>kg/m$^2$ (median, IQR) | 21.9 (20.7-23.3) | 25.6 (22.5-26.8) | 22.9 (21.6-25.7) |
| Usual daily drinking amount<br>ml (median, IQR) | 1500 (1500-2000) | 2000 (1500-2475) | 2000 (1500-2000) |
| Drinking amount at the day of US<br>ml (median, IQR) | 1000 (500-1050) | 1200 (775-1500) | 1000 (600-1275) |
| Blood pressure systolic<br>mmHg (median, IQR) | 117.5 (108.5-122) | 123.5 (115-128) | 119 (109-125) |
| Blood pressure diastolic<br>mmHg (median, IQR) | 74 (72-79) | 76 (71-81) | 76 (71-80) |
| Pulse<br>bpm (median, IQR) | 68 (61.5-76) | 67 (61-77) | 68 (61-77) |
| Alcohol occasionally<br>n (%) | 14 (93.3) | 14 (93.3) | 28 (93.3) |
| Smoking=yes n (%) | 1 (6.6) | 2 (13.3) | 3 (10) |
| Medication=yes<br>n (%) | 9 (60) | 2 (13.3) | 11 (36.7) |

Baseline characteristics of the participants, according to sex and overall. SD = standard deviation, IQR = interquartile range, bpm = beats per minute.

## 3.2. Primary endpoint

Median uJVP was 3.8 cmH₂O (IQR 3.0–4.0) for method 1, 4.5 cmH₂O (IQR 3.8–5.3) for method 2, 3.9 cmH₂O (IQR 3.3–4.5) for method 3a and 3.3 cmH₂O (IQR 2.8–3.5) for method 3b. The percentage of participants whose measurements differed by >1 SD from the expected value was 56.7%, 21.4%, 51.7% and 83.3% for methods 1-3b, respectively (Table 3, Fig 3), all below the expected value.

## 3.3. Sensitivity analysis

Although altogether male participants showed slightly higher uJVP values than females, there was no relevant differences in the longitudinal methods 3a (median 4.1 vs 3.7, p = 0.37) or 3b (median 3.4 vs 3.0 p = 0.28), whereas both transversal methods 1 (median 4.0 vs 3.1, p = 0.02) and 2 (median 5.4 vs 4.0, p = 0.01) showed a sex-related difference.

**Table 3. Primary and secondary endpoints.**

|  | Participants with uJVP>1 SD from expected measurement n/available assessment pairs (%) | Reproducibility/ inter-observer agreement n/total participants (%) | Successful visualization n/total participants (%) | Duration (s) Median (IQR) calculated for successful visualizations | Tolerability (VAS, 10=best) Median (IQR) calculated for all participants |
|---|---|---|---|---|---|
| Method 1 | 17/30 (56.7) | 18/30 (60) | 30/30 (100) | 29.5 (20.8-41.3) | 9.8 (8.6-10) |
| Method 2 | 3/14 (21.4) | 7/30 (23.3) | 14/30 (47) | 32.5 (17.0-45.0) | 9.5 (8.6-10) |
| Method 3a | 14/29 (51.7) | 19/30 (63.3) | 29/30 (97) | 38.5 (23.5-62.0) | 9.5 (9.0-10) |
| Method 3b | 25/30 (83.3) | 25/30 (83.3) | 30/30 (100) | 38.5 (23.5-62.0) | 9.5 (9.0-10) |

Results of the primary and secondary endpoints for the different methods. Primary endpoint percentages were calculated using available assessment pairs as the denominator; secondary endpoint percentages were calculated using the total number of participants. Median duration was calculated only for successful visualizations; median VAS was calculated for all participants. uJVP = ultrasound jugular venous pressure, SD = standard deviation, IQR = interquartile range, VAS = visual analogue scale.

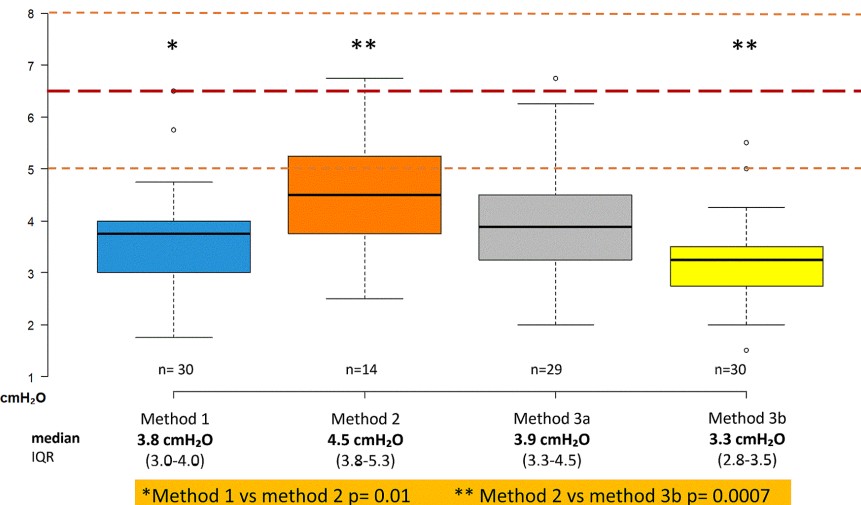

**Fig 3. Ultrasound based Jugular Venous Pressure (uJVP) results.** Box-plot showing uJVP-results according to method. y-axis: in cmH₂O; dotted lines = expected mean 6.5 cmH₂O (red) ± 1 SD (orange) based on previous studies.

### 3.4. Secondary endpoints

Method 2 yielded significantly higher uJVP values compared to method 1 (uJVP 4.5 cmH$_2$O vs 3.8 cmH$_2$O, p = 0.01) and method 3b (uJVP 4.5 cmH$_2$O vs 3.3 cmH$_2$O, p = 0.0007) (Fig 3).

Bland-Altman analysis showed that Method 3b had the smallest bias (−0.25 vs −0.30 for method 1, −0.83 for method 2, and −0.42 for method 3a) and the narrowest limits of agreement (−2.26 to 1.76), indicating the highest interrater reproducibility (Fig 4, Table 4).

ICC with absolute agreement was 0.32 (95% CI −0.031 to 0.61, p = 0.04) for method 1; 0.45 (95% CI −0.12 to 0.83, p = 0.06) for method 2; 0.130 (95% CI −0.24 to 0.46, p = 0.24) for method 3a; and 0.49 (95% CI 0.17 to 0.72, p = 0.002) for method 3b.

The number of participants whose measurement differences between the two investigators fell within 1 cm was 18/30 (60.0%), 7/30 (23.3%), 19/30 (63.3%), and 25/30 (83.3%) for Methods 1 through 3b, respectively (Table 3).

IJV-height visualization was possible in 100% of cases for methods 1 and 3b, 97% for method 3a and 47% for method 2. The median duration was 29.5 s (IQR 20.8–41.3) for method 1, 32.5 s (IQR 17.0–45.0) for method 2, and 38.5 s

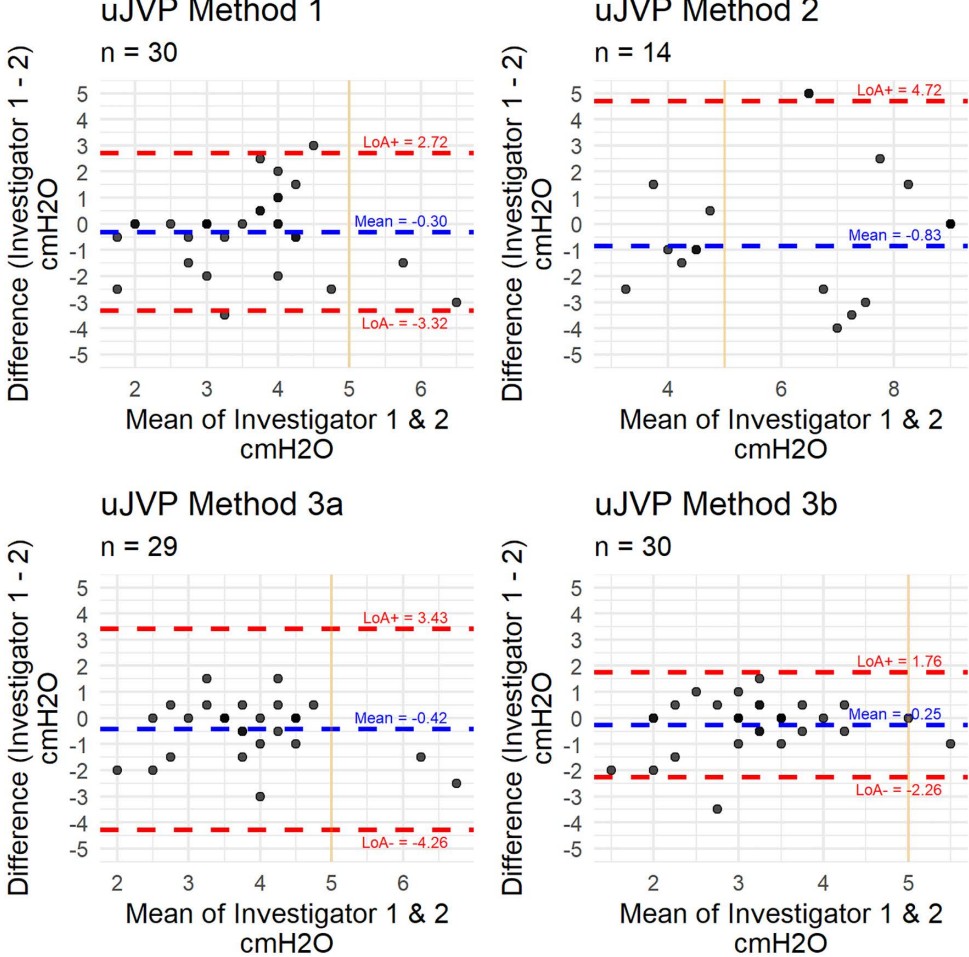

**Fig 4. Bland-Altmann analysis.** Bland-Altman plots for methods 1 to 3b. Orange vertical line identifies 5 cmH$_2$O. uJVP = ultrasound jugular venous pressure, LoA = limits of agreement.

**Table 4. Bland-Altmann analysis results.**

|  | Bias | SD | LoA lower | LoA upper |
|---|---|---|---|---|
| Method 1 | −0.30 | 1.54 | −3.32 | 2.72 |
| Method 2 | −0.83 | 2.83 | −6.38 | 4.72 |
| Method 3a | −0.42 | 1.96 | −4.26 | 3.43 |
| Method 3b | −0.25 | 1.02 | −2.26 | 1.76 |

SD= standard deviation; LoA= limit of agreement

(23.5–62.0) for method 3a/b. The median tolerability (VAS, 10 = best) was 9.8 (IQR 8.6–10.0) for method 1, 9.5 (IQR 8.6–10.0) for method 2, and 9.5 (IQR 9.0–10.0) for method 3 a/b (Table 3).

### 3.5. Serious events

No serious events occurred during the study.

### 3.6. Summary of the results

In our observational, methodological study of 30 healthy volunteers (50% female) comparing four uJVP methods, we found:

- **uJVP measurements:** All values were <8 cmH$_2$O, correctly below the hypervolemia threshold, whereas the hypovolemia threshold was exceeded in 56.7%, 21.4%, 51.7%, and 83.3% for methods 1–3b, respectively.

- **Best-performing method:** Longitudinal assessment at the base of the IJV tapering portion (method 3b) showed the highest interrater agreement (25/30, 83.3%) and could be visualized in 30/30 participants. Bland-Altmann analysis supported the use of Method 3b for consistent uJVP assessment between investigators (bias −0.25, limits of agreement −2.26 to 1.76). Acquisition time was 38.5 seconds, and tolerability (VAS 9.5) was comparable to other methods.

- **Practical recommendation:** Individualized head-of-bed elevation, visualizing uJVP in the mid-third of the neck, can help minimize confounding factors such as proximity to the subclavian vein or venous valves.

## 4. Discussion

Our study aimed to systematically compare different IJV-US methods for hydration status assessment, yielding three main findings. First, the uJVP values measured in our cohort of healthy volunteers were lower than those previously reported in the literature as normal values. Second, reproducibility varies substantially between the four applied methods, with the best results for the longitudinal assessment at the base of the tapering portion (method 3b). Third, ultrasound assessment for JVP is well tolerated by the examined person and the required duration short enough to represent an easy and applicable point-of-care test.

The first finding of our study regards the uJVP values in euvolemic patients: All the assessed methods consistently yielded uJVP measurements below the expected normal values. Those values were based on previous studies investigating and defining uJVP, amongst which there is a great heterogeneity regarding the applied ultrasound method and reference standard [25,30,35,36]. Socransky et al [35] used longitudinal US view, measuring at the base of IJV taper (equivalent to method 3b), to examine 77 supposedly euvolemic emergency department patients, and found mean uJVP to be 6.35 cm. Siva et al [30] used longitudinal and transverse US view, measuring at the tip of the tapering portion (equivalent to a combination of methods 3a/2), to correlate uJVP with invasively measured CVP in a collective of 44 patients. Wang et al [36] used transverse US view, measuring as soon as IJV was smaller than CCA (equivalent to method 1), to correlate uJVP with invasively measured RAP in a sample of 100 patients. All these studies differed from our patient

per age (mean age above 45 years old), BMI (mean BMI above 25 kg/m$^2$) and head of bed elevation (fixed at 45°); and in most cases, uJVP was observed just above the clavicle. This is the site of confluence into the subclavian vein, where the visualized tapering of IJV may be overestimated due to a) negative intrathoracic pressure splinting the vein open as it enters the chest cavity, and b) venous valves present in this region [28,37]. In our study, all participants had uJVP values <8 cmH$_2$O and were thus correctly identified as non-hypervolemic according to the described cut-offs. Therefore, we support the conclusion of a metanalysis published by Wang et al in 2022 [25] that uJVP is useful for assessing hypervolemia, but not suitable for identifying hypovolemia.

One could argue that our lower values were obtained because we did not use a fixed head-to-bed elevation angle. However, we advocate for our approach using individualized head-of-bed elevation angle, as it reduces confounding factors through anatomical influences; reason why it has recently also been applied in other studies [38]. In addition, this is a pragmatic, practice-oriented approach, as maintaining a fixed head-of-bed elevation angle is not always feasible in routine clinical settings.

The second result of our study was that longitudinal assessment at the base of the tapering portion showed the highest reproducibility, both in the Bland-Altman analysis and when using a narrower limit of agreement of 1 cm. Moreover, it showed the highest ICC when looking at absolute agreement. One possible explanation is the reduced susceptibility of this method to probe pressure, a known confounding factor for IJV-US assessment [41], in contrast to the transversal methods 1 and 2. Further, IJV layering (i.e., the effect that a non-collapsed IJV is visualized due to enhanced venous return from cranial veins, independent of right atrial pressure [37]) may alter the location of the tip (method 3a) but not the base of the tapering portion. One could argue that a reproducibility of about 83% may still be insufficient for reliable use in clinical practice. However, IVC-US, which is currently most used for US hydration status assessment, has also shown only a moderate reproducibility [38,42,43]. Importantly, longitudinal uJVP measurements showed robust results with no significant differences between male and female participants, in contrast to transverse measurements. As uJVP is an indirect measure of CVP, which shows no sex-related differences [44], no differences were expected. This again underlines the greater reliability of longitudinal measurements compared to transverse ones.

The third finding of our study concerns another key feature of a standardized method: feasibility. In our study, Method 3b was well tolerated and yielded results in 100% of participants. Additionally, the assessment duration had a median time of less than one minute, supporting previous evidence [41] suggesting IJV-US as a highly feasible tool for clinical practice. In contrast, method 2, using a transverse visualization, despite showing similar results about tolerability and assessment duration, produced results in only 47% of participants, as no complete IJV collapse could be visualized until the mandibular angle in the remaining cases.

This study has several limitations, beginning with its small sample size and observational design. However, to the best of our knowledge, it represents the largest cohort in which all four previously described methods were systematically assessed by two independent, blinded investigators. Further, we did not compare our uJVP measurements to invasive volume status assessment methods (i.e., CVP, RAP) or other proposed ultrasound evaluations (i.e., IVC-US, echocardiography), to confirm euvolemic status. However, this was consistent with the study's primary objective which was methodological in nature. Additionally, our cohort consisted of young, healthy volunteers without known medical conditions affecting volume status and without clinical signs of hypo- or hypervolemia, making it likely that participants were euvolemic. Another potential limitation is that the investigators conducting the ultrasound assessments had only basic ultrasound skills. However, all investigators underwent two structured training sessions and adhered strictly to a standardized examination protocol. Moreover, this in fact enhances the study's generalizability.

## 5. Conclusion

In conclusion, we advocate for a standardized approach to IJV-US in future studies and clinical applications. Our findings support the use of a longitudinal IJV view, with uJVP measurement taken at the base of the tapering portion. Head-of-bed

elevation angle should be adjusted individually to minimize confounding factors. Further studies are needed to correlate uJVP measured with this method to invasively assessed CVP or RAP, allowing for consistent definition of hypovolemia.

## Acknowledgments

We sincerely thank all the participants for their valuable contribution to this study. We are also deeply grateful to the dedicated staff of the outpatient day clinic for their support and assistance throughout the assessment process.

## Author contributions

**Conceptualization:** Karin Vogt, Mark Kirsch, Laura Potasso.

**Data curation:** Karin Vogt, David Widmer, Laura Potasso.

**Formal analysis:** Laura Potasso.

**Funding acquisition:** Laura Potasso.

**Investigation:** Karin Vogt, David Widmer, Mark Kirsch, Laura Potasso.

**Methodology:** Karin Vogt, David Widmer, Mark Kirsch.

**Project administration:** Karin Vogt.

**Resources:** Laura Potasso.

**Software:** Karin Vogt, Laura Potasso.

**Supervision:** Mark Kirsch, Laura Potasso.

**Validation:** Karin Vogt, David Widmer, Mark Kirsch, Laura Potasso.

**Visualization:** Laura Potasso.

**Writing – original draft:** Karin Vogt.

**Writing – review & editing:** David Widmer, Mark Kirsch, Laura Potasso.

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
