## [Decision Letter · Decision Letter 0]

1 Oct 2025

Dear Dr. Potasso,

Thank you for submitting your manuscript to PLOS ONE. After careful consideration, we feel that it has merit but does not fully meet PLOS ONE’s publication criteria as it currently stands. Therefore, we invite you to submit a revised version of the manuscript that addresses the points raised during the review process.

We look forward to receiving your revised manuscript.

Kind regards,

Vincenzo Francesco Tripodi

Academic Editor

PLOS ONE

Journal Requirements:

https://journals.plos.org/plosone/s/file?id=ba62/PLOSOne_formatting_sample_title_authors_affiliations.pdf....

“This study was funded by a research grant from the Margot and Erich Goldschmidt & Peter René Jacobson-Foundation 122023 to LP (https://www.goldschmidtjacobsonstiftung.com/). LP was the sponsor/PI playing a role in study design, data collection and analysis and manuscript writing and submission”

3. We note that your Data Availability Statement is currently as follows: [All relevant data are within the manuscript]

Additional Editor Comments (if provided):

Dear Authors,

Thank you for submitting your manuscript to our journal. The objectives and the rationale are clearly reported. The manuscript structure and writing are well-organized and easily comprehensible.

The study design and limitations of the study are well described in the text.

This manuscript requires some adjustments. I kindly invite you to examine the suggestions and advice offered, especially concerning the statistical aspects.

Best regards,

VF Tripodi

Reviewers' comments:

Reviewer's Responses to Questions

**Comments to the Author**

1. Is the manuscript technically sound, and do the data support the conclusions?

Reviewer #1: Partly

Reviewer #2: Yes

2. Has the statistical analysis been performed appropriately and rigorously?

Reviewer #1: Yes

Reviewer #2: No

3. Have the authors made all data underlying the findings in their manuscript fully available?

Reviewer #1: Yes

Reviewer #2: Yes

4. Is the manuscript presented in an intelligible fashion and written in standard English?

Reviewer #1: Yes

Reviewer #2: Yes

Reviewer #1: Hello

This is a good article

Of course, I think you can make the title of the article shorter and more concise.

The text of the article is good, but it is better to collect and present the important points of the article in one place and in a few lines.

Good luck.

Reviewer #2: The manuscript addresses an interesting and clinically relevant methodological question by assessing the reproducibility and feasibility of different ultrasound jugular venous pressure (uJVP) measurement techniques. The study is thoughtfully designed, clearly presented, and strengthened using a balanced, healthy volunteer population with blinded assessments. The authors are to be appreciated for their valuable contribution. With some refinements, this work has the potential to make a meaningful impact and be suitable for publication.

1. Correct the sequence of in-text citation references in the introduction (line 62), i.e., [1,2,12,4–11].

2. Statistical analysis (line 177 to 182)

I. The authors need to refine their choice of statistical tests. While the Wilcoxon rank-sum test is appropriate for comparing independent groups (e.g., male vs. female). Comparisons of uJVP values against expected values (one-sample data) would be more appropriately analyzed using a Wilcoxon signed-rank test. Similarly, when comparing different methods within the same patients (paired data), the Wilcoxon signed-rank test would be preferable.

Suggestion: For clarity, the authors could revise the statistical analysis section to indicate that both the Wilcoxon signed-rank test (for paired and one-sample comparisons) and the Wilcoxon rank-sum test (for independent groups) were applied as appropriate. This adjustment would improve the accuracy and transparency of the statistical analysis.

II. The authors need to reconsider how reproducibility is presented. Reporting it as the “percentage of participants with the same measurement within 1 cm” provides some indication of agreement but does not represent a standard reliability metric. More conventional approaches, such as intraclass correlation coefficient (ICC) or Bland–Altman analysis, would provide a stronger and more robust evaluation of inter-observer agreement. If these analyses are not possible, it would be helpful to explain the rationale for selecting a 1 cm tolerance and to note the potential limitations of this approach.

.

Reviewer #1: No

Reviewer #2: No

---

## [Author Response · Author response to Decision Letter 1]

22 Oct 2025

Longitudinal ultrasound jugular venous pressure assessment shows high reproducibility and feasibility to detect hypervolemia – an observational study in healthy volunteers

New title: Longitudinal ultrasound assessment of jugular venous pressure reliably detects hypervolemia: an observational study in healthy volunteers

PONE-D-25-44241

Point-by-point answer

Dear Authors,

Thank you for submitting your manuscript to our journal. The objectives and the rationale are clearly reported. The manuscript structure and writing are well-organized and easily comprehensible.

The study design and limitations of the study are well described in the text.

This manuscript requires some adjustments. I kindly invite you to examine the suggestions and advice offered, especially concerning the statistical aspects.

Best regards,

VF Tripodi

Dear Professor Tripodi,

We sincerely thank you for your thoughtful feedback and positive evaluation of our manuscript. We appreciate your acknowledgment of our study’s clarity, structure, and design.

We have carefully considered the reviewers’ and your suggestions, particularly regarding the statistical aspects, and have made the necessary revisions to improve the manuscript accordingly. Detailed responses to each comment are provided in the attached revision report. Here you find the point-by-point answers addressing all comments and suggestions.

Best regards

Laura Potasso on behalf of all authors

Review Comments to the Author

Reviewer #1: Hello

This is a good article

Of course, I think you can make the title of the article shorter and more concise.

The text of the article is good, but it is better to collect and present the important points of the article in one place and in a few lines.

Good luck.

We sincerely thank the reviewer for this helpful suggestion. Following your advice, we have shortened the title to make it more concise and focused. In addition, we have summarized the most important findings of the study in a few lines at the end of the results section to provide a clear overview of the main results.

New Title:

Longitudinal ultrasound assessment of jugular venous pressure reliably detects hypervolemia: an observational study in healthy volunteers

New paragraph (lines 244-258):

3.6 Summary of the results

In our observational, methodological study of 30 healthy volunteers (50% female) comparing four uJVP methods, we found:

• uJVP measurements: All values were <8 cmH₂O, correctly below the hypervolemia threshold, whereas the hypovolemia threshold was exceeded in 56.7%, 21.4%, 51.7%, and 83.3% for methods 1–3b, respectively.

• Best-performing method: Longitudinal assessment at the base of the IJV tapering portion (method 3b) showed the highest interrater agreement (25/30, 83.3%) and could be visualized in 30/30 participants. Bland-Altmann analysis supported the use of Method 3b for consistent uJVP assessment between investigators (bias -0.25, limits of agreement -2.26 to 1.76). Acquisition time was 38.5 seconds, and tolerability (VAS 9.5) was comparable to other methods.

• Practical recommendation: Individualized head-of-bed elevation, visualizing uJVP in the mid-third of the neck, can help minimize confounding factors such as proximity to the subclavian vein or venous valves.

Reviewer #2:

The manuscript addresses an interesting and clinically relevant methodological question by assessing the reproducibility and feasibility of different ultrasound jugular venous pressure (uJVP) measurement techniques. The study is thoughtfully designed, clearly presented, and strengthened using a balanced, healthy volunteer population with blinded assessments. The authors are to be appreciated for their valuable contribution. With some refinements, this work has the potential to make a meaningful impact and be suitable for publication.

1. Correct the sequence of in-text citation references in the introduction (line 62), i.e., [1,2,12,4–11].

We thank the reviewer for pointing at this. We updated the references accordingly.

2. Statistical analysis (line 177 to 182)

I. The authors need to refine their choice of statistical tests. While the Wilcoxon rank-sum test is appropriate for comparing independent groups (e.g., male vs. female). Comparisons of uJVP values against expected values (one-sample data) would be more appropriately analyzed using a Wilcoxon signed-rank test. Similarly, when comparing different methods within the same patients (paired data), the Wilcoxon signed-rank test would be preferable.

Suggestion: For clarity, the authors could revise the statistical analysis section to indicate that both the Wilcoxon signed-rank test (for paired and one-sample comparisons) and the Wilcoxon rank-sum test (for independent groups) were applied as appropriate. This adjustment would improve the accuracy and transparency of the statistical analysis.

Thanks for this important suggestion. We revised the method section accordingly (lines 176-180):

We compared uJVP values to the predefined threshold using a one-sample Wilcoxon signed-rank test. Differences between methods within the same participants were assessed using a paired Wilcoxon signed-rank test. Sex differences in uJVP values were evaluated using the Wilcoxon rank-sum test (Mann–Whitney U test).

II. The authors need to reconsider how reproducibility is presented. Reporting it as the “percentage of participants with the same measurement within 1 cm” provides some indication of agreement but does not represent a standard reliability metric. More conventional approaches, such as intraclass correlation coefficient (ICC) or Bland–Altman analysis, would provide a stronger and more robust evaluation of inter-observer agreement. If these analyses are not possible, it would be helpful to explain the rationale for selecting a 1 cm tolerance and to note the potential limitations of this approach.

Thank you for raising this important point. We have added the Bland-Altman analysis to better visualize and compare the results. The rationale for using the percentage-based method was that, although the Bland-Altman analysis showed that Method 3b had the narrowest limits of agreement, these limits were within ca. 2 cm, which we considered insufficiently precise for clinical use. Therefore, we retained the analysis using a 1 cm difference to quantify the proportion of participants whose measurements fell within this stricter range.

We added the Bland-Altmann analysis in the section methods (lines 183-185), results (lines 227-236) and discussion (lines 294-295) as well as Figure 4.

Figure 4 – Bland-Altmann analysis

Bland-Altman plots for methods 1 to 3b. uJVP= ultrasound jugular venous pressure, LoA= limits of agreement.

---

## [Decision Letter · Decision Letter 1]

30 Nov 2025

Dear Dr. Potasso,

Thank you for submitting your manuscript to PLOS ONE. After careful consideration, we feel that it has merit but does not fully meet PLOS ONE’s publication criteria as it currently stands. Therefore, we invite you to submit a revised version of the manuscript that addresses the points raised during the review process.

DATE_REVISION_DUE%. If you will need more time than this to complete your revisions, please reply to this message or contact the journal office at plosone@plos.org. . . . A rebuttal letter that responds to each point raised by the academic editor and reviewer(s). You should upload this letter as a separate file labeled 'Response to Reviewers'.A marked-up copy of your manuscript that highlights changes made to the original version. You should upload this as a separate file labeled 'Revised Manuscript with Track Changes'.An unmarked version of your revised paper without tracked changes. You should upload this as a separate file labeled 'Manuscript'.

We look forward to receiving your revised manuscript.

Kind regards,

Vincenzo Francesco Tripodi

Academic Editor

PLOS ONE

Journal Requirements:

Reviewers' comments:

Reviewer's Responses to Questions

**Comments to the Author**

Reviewer #3: All comments have been addressed

Reviewer #4: (No Response)

2. Is the manuscript technically sound, and do the data support the conclusions?

Reviewer #3: Yes

Reviewer #4: Yes

3. Has the statistical analysis been performed appropriately and rigorously?

Reviewer #3: Yes

Reviewer #4: Yes

4. Have the authors made all data underlying the findings in their manuscript fully available?

Reviewer #3: Yes

Reviewer #4: Yes

5. Is the manuscript presented in an intelligible fashion and written in standard English?

Reviewer #3: Yes

Reviewer #4: Yes

Reviewer #3: Dear authors,

I appreciate the opportunity to review the manuscript titled "Longitudinal ultrasound assessment of jugular venous pressure reliably detects hypervolemia: an observational study in healthy volunteers".

The manuscript structure and writing are well-organized and easily comprehensible.

The topic is truly interesting, and the manuscript has been revised in accordance with the reviewers’ requests.

Reviewer #4: • Primary endpoint choice and interpretation—be explicit and justify it

What you used: “proportion of participants whose measurements differed >1 SD from the expected uJVP value (6.5 ± 1.5 cm H₂O).”

• Reproducibility: add standard reliability metrics (ICC ± CI) and present complete Bland-Altman numbers

What you did: percent agreement within ±1 cm and Bland-Altman plots; you state method 3b bias -0.25 and LoA -2.26 to 1.76 and percentages within 1 cm for each method.

Why this is insufficient: percent within ±1 cm is informative but not a standard reliability metric; ICC (two-way random, absolute agreement) with 95% CI and a table of Bland-Altman bias and 95% limits of agreement (numerical values for all methods) are expected

• Missing/ambiguous denominators — make “n/total” consistent and transparent

Issue: method 2 has only 14 successful visualizations; Table 3 reports results for method 2 with denominators that look inconsistent (e.g., 3/14 appears in snippet). The abstract and different parts of the manuscript show slightly different percentages for method 2 (e.g., 21.4% vs 28.6% appears in different places of the document snippets). These inconsistencies must be fixed.

• Sample size calculation wording and assumptions need clarification

The sample size section says “alpha and beta error of 0.05 and a power of 95%” that wording is confusing (alpha = 0.05 is fine; beta = 0.05 corresponds to power = 95%, but writing “alpha and beta error of 0.05 and a power of 95%” is redundant/unclear). Also, the formula printed is not standardly presented (formatting/notation broken).

• Methodological suggestions

Report full Bland-Altman statistics for all methods (bias, SD, and lower/upper 95% LoA numeric values) in a table (not just plots). You already show 3b numbers; add the others.

Add ICC (with 95% CI) for each method (two-way random, absolute agreement)

Explain the rationale for 1-cm tolerance:

• Line/figure comments

• Table 3: add a footnote explaining denominators for method 2 (only 14/30 visualized) and show for duration/VAS.

• Figures 3 & 4: Label the axes with units (cmH₂O), add sample sizes to boxplots and Bland-Altman plots, and include a dashed line at the +5 cm constant if you show raw IJV height.

• Abstract: ensure the percentage reported for method 2 is the same as in the Results/Table (fix the 21.4% vs 28.6% inconsistency).

.

Reviewer #3: No

Reviewer #4: No

---

## [Author Response · Author response to Decision Letter 2]

12 Dec 2025

We have uploaded the response to the reviewer (file docx named Point-by-point_2_Vogt)

---

## [Editor Report · Decision Letter 2]

16 Mar 2026

Longitudinal ultrasound assessment of jugular venous pressure reliably detects hypervolemia: an observational study in healthy volunteers

PONE-D-25-44241R2

Dear Dr. Potasso,

We’re pleased to inform you that your manuscript has been judged scientifically suitable for publication and will be formally accepted for publication once it meets all outstanding technical requirements.

Kind regards,

Vincenzo Lionetti, M.D., PhD

Academic Editor

PLOS One
---

## [Editor Report · Acceptance letter]

PONE-D-25-44241R2

PLOS One

Dear Dr. Potasso,

I'm pleased to inform you that your manuscript has been deemed suitable for publication in PLOS One. Congratulations! Your manuscript is now being handed over to our production team.

Kind regards,

on behalf of

Prof. Vincenzo Lionetti

Academic Editor

PLOS One